# G-TIGRE: A new generative framework for Multivariate Time Series Imputation By Graph Neural Networks

## Abstract

The persistent challenge of handling missing values in multivariate time series (MTS) data demands precise solutions to avoid potential pitfalls in real-world applications. Conventional imputation methods often struggle to capture effective spatio-temporal representations of such data, failing to exploit its intrinsic temporal nature and intricate inter-variable relationships. In recent years, deep learning-based imputation methods have gained popularity. However, they often lack dedicated structures and models specifically designed to address this unique challenge. In response to these challenges, we introduce a novel framework called G-TIGRE, which synergistically leverages the capabilities of two prominent research streams in this field: Generative Adversarial Networks (GANs) and Graph Neural Networks (GNNs). GANs excel at effectively modeling data distributions, while GNNs demonstrate remarkable proficiency in extracting spatio-temporal features from data. By integrating these two techniques, which have not previously been explored together in this domain, G-TIGRE addresses several critical issues, including the elimination of the need to make assumptions about data stationarity, the ability to train with incomplete data, and the enhancement of spatio-temporal representation learning. Through extensive experiments conducted on a diverse benchmark of state-of-the-art methods, we establish that G-TIGRE achieves competitive performance, closely rivaling the top-performing models. Furthermore, an in-depth ablation study sheds light on the unique contributions of each component within G-TIGRE, elucidating its effectiveness in MTS imputation. This work introduces an exciting shift in addressing the persistent challenges of missing data in multivariate time series, with far-reaching implications across various domains.

## 1 Introduction

Data analysis frequently faces the persistent challenge of managing missing data, which can manifest due to various factors, including inadvertent errors, complications during data collection, or even deliberate omissions. This issue assumes heightened significance within the domain of time series data, particularly in the context of multivariate time series (MTS), which track multiple variables over a timeline. Consider sensors, for example, as they can malfunction or transmit incorrect data (Wu et al., 2020). Similarly, data gaps can occur in the healthcare sector due to missed tests or misplaced records (Moor et al., 2020). Furthermore, with the Internet of Things (IoT) expansion, these data concerns are escalating (Ahmed et al., 2022). Addressing these gaps in a right way is essential. Otherwise, the original data distribution is altered, affecting subsequent analyses or predictions. Due to the widespread prevalence of the problem, multivariate time series imputation (MTSI) has become a heavily researched topic of high interest.

Generative Adversarial Networks (GANs) have emerged as a prominent approach in deep learning (DL) for imputation. GANs are increasingly gaining recognition in the field, especially in their application to time series data, resulting in the development of various specialized versions. The use of generative methods holds great promise, as models based on these techniques can provide a deeper understanding of the actual distribution of MTS data. This, in turn, can significantly improve the modeling of imputation tasks (Li et al., 2023). On the other hand, there has been a recent increase in proposals calling for precise imputations using spatio-temporal methods to deal with the complexity of MTS data (Yoon et al., 2018c). These techniques use MTS's temporal dynamics and spatial attributes to capture the inter-variable relationships within the dataset effectively. To explore this further, emerging research has utilized Graph Neural Networks (GNNs) to maximize the information gained from MTS. However, these studies have not fully utilized the strengths of GANs and often require making assumptions about data stationarity (Cini et al., 2022).

This study proposes a novel model that harnesses the advantages of the two previously mentioned research paradigms. We introduce a new framework for multivariate time series imputation (MTSI): *Generative Time series Imputation with Graph-based REcurrent neural networks* (G-TIGRE). Within G-TIGRE, we exploit a Generative Adversarial Network (GAN)-based architecture, with adversarial components rooted in Graph Neural Networks (GNNs). These GAN-GNN hybrid models can bi-directionally extract spatio-temporal representations from the MTS, analyzing it both forward and backward in time. Our contributions are as follows: 1) We introduce a novel framework that enables the integration of two of the most prolific research streams in MTSI: the use of generative models and GNNs. 2) Our approach avoids assumptions about data stationarity and enables training without relying on complete real data through the use of GANs. 3) Enhancing the model's ability to capture more effective spatio-temporal representations by including GNNs. 4) Our approach achieves competitive performance comparable to the top two state-of-the-art methods. 5) We conduct an extensive ablation analysis that elucidates each component's contributions to our model's results.

This article is organized as follows: Section 2 analyses related work in the field of MTSI. In Section 3, we establish the preliminary concepts on which our method is based. Section 4 provides a detailed description of G-TIGRE . We present the results obtained in Section 5, and finally, Section 6 contains the conclusions. It is worth noting that this article includes an extensive Appendix that extends the information and presents additional results.

## 2 RELATED WORKS

### 2.1 TIME SERIES IMPUTATION

From the outset, time series imputation has been a heavily researched problem, resulting in a wide array of different methods that have been applied. These methods range from classical methods such as zero imputation, forward filling, or mean imputation, to somewhat more refined approaches like linear interpolation (Moor et al., 2020). Nevertheless, more sophisticated methods based on mathematical techniques emerged, including k-nearest neighbours (Beretta & Santaniello, 2016), linear predictors (Seaman et al., 2012), matrix factorization (MF) (Cichocki & Phan, 2009), vector autoregressive model-imputation (VAR) (Bashir & Wei, 2018), or even multivariate imputation by chained equations (MICE) (Schafer, 1997) and multivariate gaussian process (MGP) (Li & Marlin, 2016). The latter two methods can still be found in benchmark comparisons.

However, with the recent surge in DL, methods based on these techniques have been proposed for MTSI. Many of these techniques have relied on the use of recurrent neural networks (RNNs) (Suo et al., 2019; Lipton et al., 2016) due to their strong ability to extract useful representations from time series data. An example of such techniques is GRU-D (Che et al., 2018), a model that modifies the typical Gated Recurrent Units (GRU) of RNNs to introduce decay in their hidden states as they become distant from real observations.

Another notable approach is BRITS (Cao et al., 2018b), which draws inspiration from GRU-D and extends it by allowing reading bidirectional time series and extracting spatial relationships between variables within the MTS.

## 2.2 GENERATIVE MODELS FOR TIME SERIES

Generative approaches for time series data are not entirely new, as illustrated by the use of Variational Autoencoders (VAE) for MTSI (Fortuin et al., 2020). However, with the advent of GANs (Goodfellow et al., 2020), numerous adaptations of this architecture have been proposed for time series data, such as those by (Yoon et al., 2019). Within the domain of MTSI, the introduction of Generative Adversarial Imputation Networks (GAIN) Yoon et al. (2018b)), originally designed for tabular data imputation, has led to various architecture adaptations tailored to time series data. These adaptations encompass the integration of RNNs into GAIN, allowing bidirectional time series analysis (Li et al., 2023; Miao et al., 2021), and the incorporation of temporal decay mechanisms (Ni & Cao, 2022; Miao et al., 2021). These models offer the advantage of being trainable without requiring complete data availability (Yoon et al., 2018b), facilitating robust imputation, as well as their ability to improve data distribution modeling to enhance imputations (Li et al., 2023). However, it can be argued that none of them fully exploit the spatio-temporal potential due to the absence of dedicated structures such as GNNs (Cini et al., 2022).

## 2.3 GRAPH NEURAL NETWORKS FOR TIME SERIES

In recent years, there has been a surge in the application of GNNs to MTS data (Deng & Hooi, 2021; Cheng et al., 2022). However, their use in the context of MTSI has remained relatively unexplored. These architectures typically involve modifying classical DL layers to incorporate graph-related operations, enhancing their ability to extract spatial representations. Examples of such approaches include the works of Gao & Ribeiro (2022), where RNN layers are combined with Graph Convolutional Network (GCN) layers, Li et al. (2018) utilizing diffusion-convolutional networks, architectures resembling transformers like Cai et al. (2020), and methods such as Temporal Graph Networks for dynamic graphs (Rossi et al., 2020).

One notable use of GNNs for imputation with adversarial methods is presented in the work by Spinelli et al. (2020). However, this method is not specifically designed for MTSI. Recently, the most prominent work on using GNN for MTSI can be found in GRIN (Cini et al., 2022), a bidirectional model that achieves outstanding results compared to the state-of-the-art methodologies. Nevertheless, GRIN comes with the constraint of requiring stationary data assumption for operation, and, like other graph-based methods, it does not fully exploit the advantages of adversarial training.

## 3 PRELIMINARIES

### 3.1 MULTIVARIATE IMPUTATION

An MTS has $N_t$ variables or channels at each time step t. A MTS can be represented as $\boldsymbol{X}_t \in \mathbb{R}^{N_t \times d}$, which is the variable or attribute matrix whose $i$-th row contains the $d$-dimensional vector $\boldsymbol{x}_t^i \in \mathbb{R}^d$ associated with the $i$-th variable at the time-step $t$.

To model the presence of missing values, we considered a binary mask matrix $\boldsymbol{M}_t \in \{0,1\}^{N_t \times d}$, where each row $\boldsymbol{m}_t^i$ indicates which of the corresponding variables $\boldsymbol{x}_t^i$ are available in $\boldsymbol{X}_t$. It follows that $m_t^{i,j} = 0$ implies $x_t^{i,j}$ to be missing; conversely, if $m_t^{i,j} = 1$, then $x_t^{i,j}$ stores the actual sensor reading or variable measure. We denote by $\widetilde{\boldsymbol{X}_t}$ the unknown ground truth variable-measure matrix, i.e., the complete time series without missing data; $\widehat{\boldsymbol{X}_t}$ represents the imputed time series $\boldsymbol{X}_t$, where missing values have been filled using the predictions of a model; and $\overline{\boldsymbol{M}}_t$ is the binary complement of $\boldsymbol{M}_t$.

The objective of MTSI is to impute missing values in a sequence of input data, i.e., the objective is from $\widehat{\boldsymbol{X}}_t$ build $\widehat{\boldsymbol{X}}_t$ and we can define the missing data reconstruction error as Equation 1.

$$\mathcal{L}(\widehat{\boldsymbol{X}}_{[t,t+T]}, \widetilde{\boldsymbol{X}}_{[t,t+T]}, \overline{\boldsymbol{M}}_{[t,t+T]}) = \sum_{h=t}^{t+T} \sum_{i}^{d} \frac{\overline{\boldsymbol{m}}_h^i \odot \ell(\widehat{\boldsymbol{x}}_h^i, \widetilde{\boldsymbol{x}}_h^i)}{\overline{\boldsymbol{m}}_h^i \odot \overline{\boldsymbol{m}}_h^i} \tag{1}$$

where $\widehat{\boldsymbol{x}}_h^i$ is the reconstructed $\widetilde{\boldsymbol{x}}_h^i$; $h$ denotes a time window defined between $[t, t + T]$, being $T$ the size of the window; $\overline{\boldsymbol{M}}_t$ is the binary complement of the mask matrix $\boldsymbol{M}_t$; $\ell(\cdot, \cdot)$ is an element-wise error function, such as the mean square error; and $\odot$ denotes the standard dot product.

Finally, our approach imputes missing values in the context of i.i.d., and our input dataset contains real measurements.

### 3.2 Sequences of Graphs

To work with graph-based recurrent models, we have followed the same structure proposed in Cini et al. (2022). In this approach, time series data is regarded as a sequence of directed graphs, a concept commonly referred to as a *sequential representation*, where we observe a graph $\mathcal{G}_t$ with $N_t$ at each time step $t$. The graph is defined by $\mathcal{G}_t = \langle \boldsymbol{X}_t, \boldsymbol{W}_t \rangle$, where $\boldsymbol{X}_t \in \mathbb{R}^{N_t \times d}$ is the matrix containing attribute vectors $\boldsymbol{x}_t^i \in \mathbb{R}^d$; $\boldsymbol{W}_t \in \mathbb{R}^{N_t \times N_t}$ represents the adjacency matrix for each time step, and contains the weight of the edge (if any) connecting the $i$-th and $j$-th variable (node). In this work, we have made the design choice of assuming that all $\mathcal{G}_t$ share the same topology and edge weights, thus $\boldsymbol{W}_t$ and $N_t$ are constant, i.e., at each time step $\boldsymbol{W}_t = \boldsymbol{W}$ and $N_t = N$.

### 3.3 Graph-recursive Neural Networks

As demonstrated in Cini et al. (2022), the use of temporal graph-based models applied to time series can offer several advantages. These include enhanced expressiveness and improved ability to capture spatio-temporal relationships, where spatial information refers to the connections among different variables.

In the field of temporal graph models, there is a wide variety of model types depending on how they differ in their approach to extracting data representations. However, we have adopted the *time-then-graph* design. As reported in Gao & Ribeiro (2022), where the authors highlight its ability to attain superior representations compared to other approaches, this design advocates a two-step approach: first encoding the temporal information and subsequently encoding the spatial information. The process unfolds as follows:

1. **Encoding Temporal Information**: A block composed of RNN layers is employed to extract temporal information from the time series. This block encodes the information from the nodes and edges into a latent graph representation.

2. **Encoding Spatial Information**: Once a single latent graph that encodes temporal information is obtained, a GCN layer block is applied. This block uses convolutions to encode the relationships among different variables.

In this article, we have used an implementation of such models, which can be found in Torch spatio-temporal library (Cini & Marisca, 2022). Specifically, we have utilized the RNNEncGCNDecModel (RNNGCN), a model that encodes vectors using an RNN and subsequently decodes them using GCN layers, and the GRUGCNModel (GRUGCN), a model that employs a GRU as the encoder and GCNs as the decoder. This implementation allows us to explore various parameters, as outlined in Appendix A.2. For more details on implementation, see Appendix A.4.

# 4 G-TIGRE

*Generative Time series Imputation with Graph-based REcurrent neural networks* (G-TIGRE) is a framework that aims to update the design of GAIN (Yoon et al., 2018b) for its application to MTS data using GNN. Therefore, G-TIGRE initially presents a design closely similar to that of GAIN. In the following subsections, we will explore the detailed description of each G-TIGRE architectural component.

## 4.1 GAN ARCHITECTURE

Upon examining Figure 1, the most obvious observation is that G-TIGRE follows a design based on the GAN architecture. G-TIGRE is composed of two competing models: the Generator (G) and the Discriminator (D).

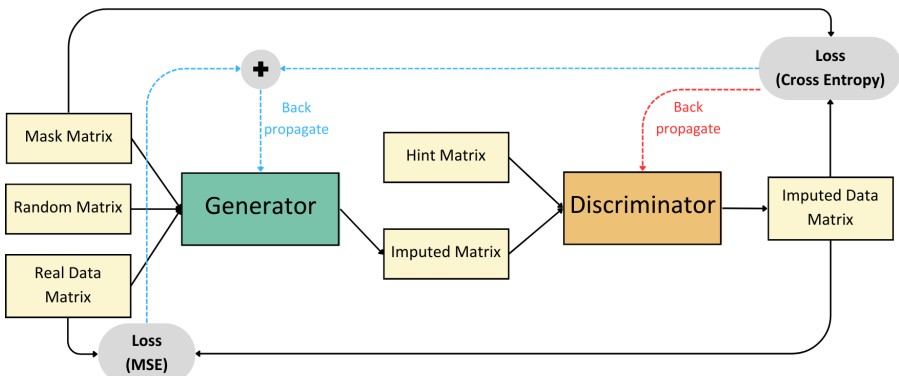

Figure 1: Visual representation of G-TIGRE 's GAN architecture.

During training, both models have distinct and opposing objectives. The goal of G is to learn how to properly fill the missing values in the original data, while D aims to identify the values generated by G. Consequently, G generates $\widehat{\boldsymbol{X}}_t$ from the matrices $\boldsymbol{X}_t$, $\widehat{\boldsymbol{M}}_t$, and a random noise matrix $\boldsymbol{Z}$, while D learns to predict $\widehat{\boldsymbol{M}}_t$ based on $\widehat{\boldsymbol{X}}_t$ and a hint matrix $\boldsymbol{H}_t$. Here, $\boldsymbol{H}_t$ is a novel element introduced by Yoon et al. (2018b) to ensure the convergence of D. For further information on G, D, and this hint matrix, refer to Appendix A.1.

Similarly to the original GAN article (Goodfellow et al., 2020) and the GAIN article (Yoon et al., 2018b), the optimization of these two models is performed through an iterative process, where G and D are modeled as temporal GNNs, as described in Section 3.3. During this iterative process, either G or D weights are updated at each step, alternating between them during training steps composed of mini-batches. Please note that $\boldsymbol{m}$, $\widehat{\boldsymbol{m}}$, and $\boldsymbol{b}$ are variables that have been defined in Appendix A.1.

$$\mathcal{L}(\boldsymbol{m}, \widehat{\boldsymbol{m}}, \boldsymbol{b}) = \sum_{i:b_i=0} \left[ m \log\left(\widehat{m}\right) + (1-m) \log\left(1-\widehat{m}\right) \right] \tag{2}$$

However, as in the last citation, the objective function to be minimized by G differs slightly from Equation 11. In this case, the specific objective function utilized is presented in Equation 3.

$$\mathcal{L}_G(\boldsymbol{m}, \widehat{\boldsymbol{m}}, \boldsymbol{b}) = - \sum_{i:b_i=0} [1 - m_i \log(\widehat{m}_i)] \tag{3a}$$

$$\mathcal{L}_M(\boldsymbol{x}, \boldsymbol{x}') = \sum_{i=1}^{d} m_i (x_i - x_i')^2 \tag{3b}$$

$$\min_G \sum_{j=1}^{K} \mathcal{L}_\mathcal{G}(\boldsymbol{m}(j), \widehat{\boldsymbol{m}}(j), \boldsymbol{b}(j)) + \alpha \mathcal{L}_M(\boldsymbol{x}(j), \widehat{\boldsymbol{x}}(j)) \tag{3c}$$

Where $\mathcal{L}_G$ is the function that evaluates the performance of G in deceiving D, and $\mathcal{L}_M$ is the loss function indicating the reconstruction error. $\mathcal{L}_M$ is used in Yoon et al. (2018b) to enable G to learn to generate imputations as accurately as possible. Finally, $\alpha$ is a hyperparameter that indicates the weight of $\mathcal{L}_M$ in the final cost function.

## 4.2 BIDIRECTIONAL APPROACH

G-TIGRE allows extracting time series representations by reading them forward and backward before proceeding with imputation, as proposed in Ni & Cao (2022).

In order to implement G-TIGRE within this framework, both D and G follow a transformation into two identical modules: one to extract representations in a forward temporal order and the other to extract representations in a backward temporal order. Subsequently, this information is fused through a final ensemble incorporating a Multi-Layer Perceptron (MLP). The alteration in G's computation is depicted in Equation 4, in addition to being visually represented in Figure 2.

$$\boldsymbol{S}_f = G_f(\boldsymbol{X}_f, \boldsymbol{M}_f, (\overline{\boldsymbol{M}}_f \odot \boldsymbol{Z}_f)) \tag{4a}$$

$$\boldsymbol{S}_b = G_b(\boldsymbol{X}_b, \boldsymbol{M}_b, (\overline{\boldsymbol{M}}_b \odot \boldsymbol{Z}_b)) \tag{4b}$$

$$\overline{\boldsymbol{X}} = MLP([\boldsymbol{S}_f||\boldsymbol{S}_b]) \tag{4c}$$

In Equation 4, $\boldsymbol{S}_f$ and $\boldsymbol{S}_b$ represent the spatio-temporal representations extracted by $G_f$ and $G_b$, respectively. $\boldsymbol{X}_f, \boldsymbol{X}_b, \boldsymbol{M}_f, \boldsymbol{M}_b$ are the data matrices and masks read in the forward and backward directions, respectively. $\boldsymbol{Z}_f$ and $\boldsymbol{Z}_b$ denote two different noise matrices. Note that the subscript $t$ is omitted for legibility when referring to these matrices, and $\overline{\boldsymbol{X}}$ is an intermediate imputation matrix already presented in Appendix A.1; finally, $\odot$ is the standard dot product. Similarly, the change in D can be observed in Equation 5. As shown, both models undergo a similar change, where $\widehat{\boldsymbol{X}}_f, \boldsymbol{H}_f, \widehat{\boldsymbol{X}}_b$, and $\boldsymbol{H}_b$ represent the imputed data matrices and the hint matrices in forward and backward representation, respectively. Once again, $\boldsymbol{S}_f$ and $\boldsymbol{S}_b$ denote the extracted spatio-temporal representations.

$$\boldsymbol{S}_f = D_f(\widehat{\boldsymbol{X}}_f, \boldsymbol{H}_f) \tag{5a}$$

$$\boldsymbol{S}_b = D_b(\widehat{\boldsymbol{X}}_b, \boldsymbol{H}_b) \tag{5b}$$

$$\widehat{\boldsymbol{M}} = MLP([\boldsymbol{S}_f||\boldsymbol{S}_b]) \tag{5c}$$

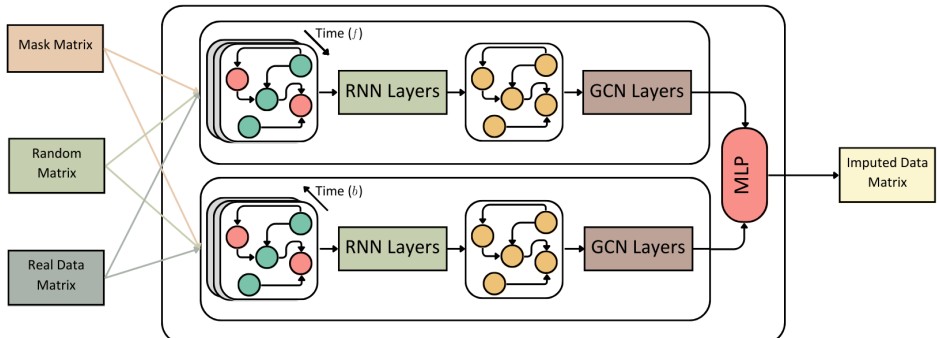

Figure 2: In this illustration, the inner workings of G are depicted. It demonstrates how its bidirectional approach passes data through two modules that apply Temporal GNNs in different temporal directions. As described in Section 3.3, these modules first extract temporal information into a latent graph, to which the GCN block is subsequently applied. Finally, the representations extracted by the two modules are combined through an MLP to perform the final imputation.

### 4.3 MULTIPLE IMPUTATION

Given that G utilizes a noise vector $z$ sampled from a random distribution, there is a possibility that, due to their inherent randomness, the generation of these values may settle in suboptimal regions of the latent space, thereby complicating the accurate imputation of values by G. To address this issue, we have chosen to implement multiple imputation, as introduced in Li et al. (2023).

This multiple imputation approach involves an iterative process, performed $n$ times, once G and D have already been trained. During each iteration, different $\boldsymbol{Z}$ matrices are generated. In each step, we also obtain $\widehat{\boldsymbol{X}}_j = G(\boldsymbol{X}, \boldsymbol{Z}_j)$ and $\widehat{\boldsymbol{M}}_j = D(\widehat{\boldsymbol{X}}, \boldsymbol{H}_j)$, where $j$ indicates the current iteration. Finally, after obtaining various possible imputations from G, we use D as an evaluator of imputation quality. For each value to be imputed, we select the one whose probability of being real is the highest according to D's predictions.

## 5 EMPIRICAL EVALUATION

In this section, we present the results obtained through experimental testing of the proposed model on two distinct datasets. We will conduct comparisons with established state-of-the-art models and analyse how the various components of the proposed model contribute to enhancing the results. Our findings reveal that our novel approach yields results on par with the state of the art, allowing for increased flexibility compared to the models with which it is compared. Furthermore, this new approach, which combines generative and graph-based models for imputing MTS, represents a previously unexplored avenue in this context.

### 5.1 DATASETS

The experimentation in this study involves two primary datasets, namely **PEMS-BAY** and **METR-LA**, both originally introduced by Li et al. (2018). These datasets provide valuable insights into traffic patterns within distinct regions, leveraging sensor measurements distributed throughout these areas. To ensure the comparability of our results with those presented in Cini et al. (2022), we have meticulously followed the same data preprocessing procedures employed in their work. This consistency extends to the generation of the adjacency matrix, which relies on the methodologies proposed by Li et al. (2018) and Wu et al. (2019).

The **PEMS-BAY** dataset focuses on traffic data within the San Francisco Bay Area. It is structured as a directed graph comprising 325 nodes and 2369 edges. Initially, this dataset exhibits a missing value rate of 0.02%. However, following the preprocessing steps, the percentage of missing values increases to 25%.

Conversely, the **METR-LA** dataset focuses on traffic data pertaining to Los Angeles County Highways. Similarly, it adopts a directed graph format comprising 207 nodes and 1515 edges. In its initial form, this dataset contains an 8.1% rate of missing values. Nevertheless, after the prescribed preprocessing, the dataset concludes with a 23% rate of missing values. Please refer to Appendix A.5 for more comprehensive details on these data sets.

## 5.2 RESULTS

Having established that our datasets possess the same characteristics as those used in Cini et al. (2022), we will proceed to ensure the comparability of results by evaluating them using the metrics of mean absolute error (MAE), mean square error (MSE), and mean relative error (MRE) (For more details about these metrics, refer to Appendix A.3). With confidence in the comparability of our results with those in the last mentioned article, Table 1 presents a comparative analysis of the outcomes achieved in this article.

Table 1: Results achieved by G-TIGRE compared with the experimentation conducted in Cini et al. (2022). As in the original article, G-TIGRE 's performance is averaged over five runs. All the methods included in the benchmark have been previously introduced in Section 2.

| Models | PEMS-BAY | | | METR-LA | | |
|---|---|---|---|---|---|---|
| | MAE | MSE | MRE (%) | MAE | MSE | MRE (%) |
| Mean | $5.42 \pm 0.00$ | $86.59 \pm 0.00$ | $8.67 \pm 0.00$ | $7.56 \pm 0.00$ | $142.22 \pm 0.00$ | $13.10 \pm 0.00$ |
| KNN | $4.30 \pm 0.00$ | $49.80 \pm 0.00$ | $6.90 \pm 0.00$ | $7.88 \pm 0.00$ | $129.29 \pm 0.00$ | $13.65 \pm 0.00$ |
| MF | $3.29 \pm 0.01$ | $51.39 \pm 0.64$ | $5.27 \pm 0.02$ | $5.56 \pm 0.03$ | $113.46 \pm 1.08$ | $9.62 \pm 0.05$ |
| MICE | $3.09 \pm 0.02$ | $31.43 \pm 0.41$ | $4.95 \pm 0.02$ | $4.42 \pm 0.07$ | $55.07 \pm 1.46$ | $7.65 \pm 0.12$ |
| VAR | $1.30 \pm 0.00$ | $6.52 \pm 0.01$ | $2.07 \pm 0.01$ | $2.69 \pm 0.00$ | $21.10 \pm 0.02$ | $4.66 \pm 0.00$ |
| rGAIN | $1.88 \pm 0.02$ | $10.37 \pm 0.20$ | $3.01 \pm 0.04$ | $2.83 \pm 0.01$ | $20.03 \pm 0.9$ | $4.91 \pm 0.01$ |
| BRITS | $1.47 \pm 0.00$ | $7.94 \pm 0.03$ | $2.36 \pm 0.00$ | $2.34 \pm 0.00$ | $16.46 \pm 0.05$ | $4.05 \pm 0.00$ |
| MPGRU | $1.11 \pm 0.00$ | $7.59 \pm 0.02$ | $1.77 \pm 0.00$ | $2.44 \pm 0.00$ | $22.17 \pm 0.03$ | $4.22 \pm 0.00$ |
| **GRIN** | $\mathbf{0.67} \pm \mathbf{0.00}$ | $\mathbf{1.55} \pm \mathbf{0.01}$ | $\mathbf{1.08} \pm \mathbf{0.00}$ | $\mathbf{1.91} \pm \mathbf{0.00}$ | $\mathbf{10.41} \pm \mathbf{0.03}$ | $\mathbf{3.30} \pm \mathbf{0.00}$ |
| G-TIGRE | $0.94 \pm 0.06$ | $2.43 \pm 0.13$ | $1.73 \pm 0.1$ | $2.38 \pm 0.03$ | $15.65 \pm 0.33$ | $5.22 \pm 0.07$ |

In Table 1, it is evident that the outcomes produced by our model fall between those of MPGRU and GRIN. We consider this a favorable result, considering that we are introducing a novel methodology. This new approach allows us to relax the stationarity constraint of GRIN while transitioning from a purely supervised training paradigm to an adversarial one.

To gain insights into the contributions of different components in improving G-TIGRE 's performance, we conducted model ablations and analyzed their impact on imputations; the results of these ablations can be seen in Table 2. Removing bidirectional analysis revealed its benefits in METR-LA; however, in PEMS-BAY, this change caused a marginal 0.01 reduction in MAE with a slight increase in standard deviation. Consequently, we can deduce that this model component can enhance performance in specific datasets, but its inclusion does not yield detrimental effects in cases where it does not. Conversely, omitting multiple imputations consistently led to degraded results, highlighting the significance of initial values in the random noise matrices.

We also performed ablation experiments on G's cost function, consisting of two vital components: $\mathcal{L}_G$ (adversarial training loss) and $\mathcal{L}_M$ (reconstruction loss encouraging G to learn data distribution). These experiments, whose results are summarized in Table 2, involved removing both $\mathcal{L}_G$ and $\mathcal{L}_M$. Removal of $\mathcal{L}_G$ leads to G being trained as an autoencoder, resulting in inferior results. This change caused G to lose its ability to capture essential data representations unattainable with a traditional loss function (Lotter et al., 2015). In contrast, eliminating $\mathcal{L}_M$ hindered the convergence of the model.

Finally, our attention turns to conducting graph ablations to assess their impact on the final results. It is important to note that these findings provide insights into the contributions of the initial recurrent network blocks in our models compared to the influence of the GCN

Table 2: Results from all types of ablations conducted to G-TIGRE . The results are averaged over 5 iterations and expressed in terms of MAE.

|  | PEMS-BAY | METR-LA |
|---|---|---|
| **G-TIGRE** | $0.94 \pm \textbf{0.06}$ | $\textbf{2.38} \pm \textbf{0.02}$ |
| w/o bi-directional | $\textbf{0.93} \pm 0.07$ | $2.45 \pm 0.03$ |
| w/o loop | $0.98 \pm 0.09$ | $2.45 \pm 0.05$ |
| w/o $\mathcal{L}_G$ | $1.07 \pm 0.26$ | $2.40 \pm 0.02$ |
| w/o $\mathcal{L}_M$ | $21.19 \pm 16.4$ | $26.08 \pm 16.44$ |
| fully connected | $0.99 \pm 0.06$ | $2.4 \pm 0.02$ |
| no edges | $1.05 \pm 0.17$ | $2.41 \pm 0.24$ |

blocks. In this context, we conducted two distinct ablations. The first involved creating a fully connected graph where all nodes were interconnected, introducing significant noise into the graph's topology. The second experiment involved the construction of an isolated graph in which all nodes remained disconnected, effectively eliminating potential benefits of graph-based structures. Upon reviewing the results presented in Table 2, it becomes evident that both ablations lead to suboptimal results, with the poorest performance observed when employing an empty graph. These findings emphasize the advantages of using graph-based models and underscore the critical importance of effective graph topology design.

Furthermore, in Appendix A.6, we present the results of a sensitivity analysis, where we evaluate the performance of G-TIGRE under varying degrees of missing values.

## 6    CONCLUSIONS

We present G-TIGRE , an innovative approach to Multivariate Time Series Imputation (MTSI) that leverages and combines the advantages of modern techniques used in Graph Neural Networks (GNNs) and Generative Adversarial Networks (GANs). The spatio-temporal nature of MTS is exploited by GNNs, while GANs' capabilities in modeling data distribution and relaxing constraints imposed by state-of-the-art models are harnessed by G-TIGRE. Our method enables us to train models without access to the complete real data and eliminates the prerequisite of data stationarity.

To validate the effectiveness of G-TIGRE, we conducted an extensive benchmark comparison, evaluating it against nine state-of-the-art models found in the literature using two different datasets. Our results indicate that G-TIGRE not only rivals the best-performing methods but also demonstrates enhanced flexibility. Additionally, we conducted an ablation study to identify the contributions of each component within our model.

Our future research will explore the effectiveness of our proposal in various problem settings and scenarios, and we also believe that our model shows great potential for use in streaming environments. We also suggest that a more comprehensive study and design of the GNNs at the core of our model could significantly improve its capabilities. The success of our approach in MTSI suggests that it can be applied to other related problems, and we encourage further research in this direction.

REPRODUCIBILITY STATEMENT

Throughout this work, we have made every effort to ensure code accessibility and transparency. To this end, we have extended as much information as possible in the Appendix to clarify the procedures followed. For instance, in Appendix A.2, we provide details on how hyperparameter optimization was conducted. In Appendix A.3, we present the formulas for the metrics used, despite their widespread familiarity with the field. In Appendix A.4, we offer insights into the tools employed for implementation and provide information on how the tensors were managed internally within this GNN-based temporal approach. Finally, in Appendix A.5, we detail more on the datasets used.

Additionally, we have created a GitHub repository where one can find all the code and steps necessary to replicate the results of this article: **[GitHub Repository Link - Censored for Peer Review - Code available in Supplementary materials]**.

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

# A  APPENDIX

## A.1  EXTENDED GAN ARCHITECTURE INFORMATION

### A.1.1  GENERATOR

The input parameters of G are $\boldsymbol{X}$ and $\boldsymbol{M}$, and it aims to learn to generate $\widehat{\boldsymbol{X}}$, which implies learning $P(\widetilde{\boldsymbol{X}}|\boldsymbol{X})$. Additionally, a random noise matrix matrix matrix $\boldsymbol{Z} \in [0,1]^{N_t \times d}$ needs to be generated. There are various proposals in the literature regarding the distribution that $\boldsymbol{Z}$ should follow. However, in this work, we have adopted the approach from Yoon et al. (2018a), which states that $\boldsymbol{Z} \sim \mathcal{U}(0, 0.01)$.

Once the input parameters are defined, the functioning of G is described by Equations 6 and 7.

$$\overline{\boldsymbol{X}} = G(\boldsymbol{X}, \boldsymbol{M}, (\overline{\boldsymbol{M}} \odot \boldsymbol{Z})) \tag{6}$$

$$\widehat{\boldsymbol{X}} = \boldsymbol{M} \odot \boldsymbol{X} + \overline{\boldsymbol{M}} \odot \overline{\boldsymbol{X}} \tag{7}$$

$\overline{\boldsymbol{X}}$ is a matrix that contains all the values generated by G, while $\widehat{\boldsymbol{X}}$, as shown in Equation 8, is a matrix that includes both the known and the imputed values.

$$\widehat{x}_t^i = \begin{cases} \widetilde{x}_t^i & \text{if } m_t^i = 1 \\ \overline{x}_t^i & \text{if } m_t^i = 0 \end{cases} \tag{8}$$

### A.1.2  DISCRIMINATOR

The role of D is to confront G and determine which elements of $\widehat{\boldsymbol{X}}$ have been observed and which have been generated, or in other words, predict $\boldsymbol{M}$. To achieve this, D takes a hint matrix $\boldsymbol{H}$ as input parameter, which will be further detailed in the next section, and $\widehat{\boldsymbol{X}}$. Specifically, the operation of D can be represented by Equation 9. Where $\widehat{\boldsymbol{M}} \in \{0,1\}^{N_t \times d}$ is the matrix that indicates the probability of each value being real or generated.

$$\widehat{M} = D(\widehat{\boldsymbol{X}}, \boldsymbol{H}) \tag{9}$$

### A.1.3  HINT MATRIX

The hint matrix $\boldsymbol{H} \in \{0, 0.5, 1\}^{N_t \times d}$ is a novel contribution from Yoon et al. (2018b).Their theoretical analysis underscores the essential role of this element in ensuring the convergence of the algorithm. This necessity arises from the fact that, in the absence of assistance to D in predicting $\widehat{\boldsymbol{M}}$, G can adopt multiple distributions to consistently deceive D. $\boldsymbol{H}$ can be defined in various ways, and by specifying distributions such as $\boldsymbol{H}|\boldsymbol{M} = m$, we can control the amount of information we provide to D.

In Yoon et al. (2018b), a hint generation method is also proposed, accompanied by a mathematical analysis that guarantees D's proper functioning. However, for this article, we have utilized the definition established in Yoon et al. (2018a), where the original author introduces a slight variation that shows similar performance through experimental results.

Specifically, to generate $\boldsymbol{H}$, we create $\boldsymbol{B} \in [0,1]^{N_t \times d}$, a matrix of random variables with values $\boldsymbol{B} \sim \mathcal{U}(0,1)$. Once $\boldsymbol{B}$ is determined, $\boldsymbol{H}$ is defined as shown in Equation 10, where $\gamma \in [0,1]$ is the threshold value of the hint rate.

$$h_t^i = \begin{cases} m_t^i & \text{if } b_t^i < \gamma \\ 0 & \text{otherwise} \end{cases} \tag{10}$$

With this definition, we constrain $\boldsymbol{H} \in \{0, 1\}$, where the meaning of each value in the hint matrix for D is determined as follows: if $h_t^i = 1$, it represents a known real value; if $h_t^i = 0$, it is a value to be determined.

### A.1.4 OBJECTIVE

The G network is trained to maximize the probability of deceiving the D network, while D aims to maximize the probability of detecting values generated by G. This objective is represented by the function in Equation 11. The loss function employed is the log-likelihood, defined in Equation 12.

$$\min_G \max_D \mathbb{E}[\mathcal{L}(\boldsymbol{M}, \widehat{\boldsymbol{M}})] \tag{11}$$

$$\mathcal{L}(\boldsymbol{a}, \boldsymbol{b}) = \sum_{i=0}^{d} [a_i \log(b_i) + (1 - a_i) \log(1 - b_i)] \tag{12}$$

By optimizing Equation 11, the goal is for G to learn $P(\widetilde{\boldsymbol{X}}|\boldsymbol{X})$, thus achieving imputations that minimize the function in Equation 1.

### A.2 HYPERPARAMETERS OPTIMIZATION

For hyperparameter optimization, certain values were chosen based on recommendations from previous articles, while others were determined through a random search with 100 runs for each dataset and model. The established values are as follows, which took as a guide the values stipulated in Yoon et al. (2018a):

- $\alpha = 100$: This parameter is employed in Equation 3.
- $\gamma = 0.9$: This hyperparameter is utilized in Equation 10.
- $\boldsymbol{Z} \sim \mathcal{U}(0, 0.01)$: As discussed in Section A.1.1.
- Normalization method: The min-max normalization method is used, which sets all values in the range $[0, 1]$.
- Batch size $= 64$: Although the original reference specified this value as 128, it was reduced to 64 for the experimentation due to GPU memory limitations.

Furthermore, the search space used for the remaining hyperparameters is detailed in Table 3, which includes the general hyperparameters of the G-TIGRE architecture and the specific hyperparameters of the RNNGCN and GRUGCN models. This table shows that for exploring some hyperparameters with continuous values, the values to be tested are generated from a uniform distribution (Goodfellow et al., 2016).

Finally, Table 4 displays the results obtained after conducting the search. For each dataset, it showcases the chosen graph-based model along with the optimal values for all hyperparameters.

### A.3 METRICS

The metrics employed in this section are the mean absolute error (MAE), mean square error (MSE), and the mean relative error (MRE; Cao et al. (2018a)). The formulas for these metrics can be found defined in Equations 13, 14, and 15, respectively.

Table 3: Hyperparameter search for G-TIGRE using a RandomSearch with 100 runs for each dataset and model.

| model | parameters | values |
|---|---|---|
| G-TIGRE | learning rate
activation
mlp layers | $10^{\mathcal{U}(-4,-2)}$
relu, tanh, silu
1, 2, 3, 4 |
| RNNGCN | rnn layers
gcn layers
cell type
rnn dropout
gcn dropout
hidden space | 1, 2, 3
1, 2, 3
gru, lstm
$\mathcal{U}(0, 0.5)$
$\mathcal{U}(0, 0.5)$
$\lfloor T \times \mathcal{U}(0.5, 2) \rfloor$ |
| GRUGCN | norm
enc layers
gcn layers
hidden space | mean, gcn, asym, none
1, 2, 3
1, 2, 3
$\lfloor T \times \mathcal{U}(0.5, 2) \rfloor$ |

Table 4: Optimal hyperparameters found

| PEMS-BAY | | | METR-LA | | |
|---|---|---|---|---|---|
| selected model | parameters | values | selected model | parameters | values |
| GRUGCN | learning rate
activation
mlp layers
norm
enc layers
gcn layers
hidden space | $8.47 * 10^{-3}$
silu
1
gcn
2
1
38 | RNNGCN | learning rate
activation
mlp layers
rnn layers
gcn layers
cell type
rnn dropout
gcn dropout
hidden space | $2.15 * 10^{-3}$
tanh
2
3
1
lstm
0.48
0.35
26 |

$$\text{MAE}(\mathbf{y}, \widehat{\mathbf{y}}) = \frac{1}{n} \sum_{i=1}^{n} |y_i - \widehat{y}_i| \tag{13}$$

$$\text{MSE}(\mathbf{y}, \widehat{\mathbf{y}}) = \frac{1}{n} \sum_{i=1}^{n} (y_i - \widehat{y}_i)^2 \tag{14}$$

$$\text{MRE}(\mathbf{y}, \widehat{\mathbf{y}}) = \frac{1}{\sum_{i=1}^{n} y_i} \sum_{i=1}^{n} |y_i - \widehat{y}_i| \tag{15}$$

## A.4 IMPLEMENTATION DETAILS

The entire code has been implemented using Python (Van Rossum & Drake, 2009) and relies on the following open-source libraries:

- Pytorch (Paszke et al., 2017).
- Pytorch Lightning (Falcon & The PyTorch Lightning team, 2019)
- Numpy (Harris et al., 2020).
- Torch spatio-temporal (Cini & Marisca, 2022).
- Pandas (pandas development team, 2020; Wes McKinney, 2010).

Additionally, to streamline replication, a Docker image and container (Merkel, 2014) have been created for easier code execution.

Regarding the G-TIGRE algorithm, the training pseudocode is practically identical to the one described in Yoon et al. (2018b); however, there is a particular effort in handling tensor dimensions due to working with temporal graph models. When dealing with MTS, matrices are typically represented as 3-dimensional tensors with a size of $I \times T \times F$, where $I$ represents each element in the mini-batch, $T$ represents the elements in the time window, and $F$ represents the values of the different feature or variables comprising the time series. However, working with temporal graph models involves transforming our data into a 4-dimensional scheme, such as $I \times T \times N \times F$, where $N$ represents each node in the graph, acting as each individual variable, and now $F$ represents the features that compose a node.

Finally, in terms of hardware, all code execution was carried out on a PC running Ubuntu 22.04.2 LTS with the following specifications: AMD Ryzen Threadripper PRO 3955WX 16-Cores CPU, NVIDIA RTX A5000 24 GB GPU, and 8X16 GB (128GB) DDR4 RAM.

## A.5 EXTENDED DATASETS INFORMATION

This section provides extended information about the datasets as a complement to what was described in Section 5.1. Tables 5 and 6 present additional details extracted from Cini et al. (2022) regarding the graph structures of the datasets and the distribution of missing values.

Table 5: Information about the graph topology of each dataset

| Dataset | Graph | | | N. neighbors | | |
|---|---|---|---|---|---|---|
| | type | nodes | edges | mean | median | isolated nodes |
| PEMS-BAY | directed | 325 | 2369 | 7.29 | 7.0 | 12 |
| METR-LA | directed | 207 | 1515 | 7.32 | 7.0 | 5 |

Table 6: Information about the distribution of missing values in each dataset

| Dataset | Original data | | | Injected faults | | |
|---|---|---|---|---|---|---|
| | % missing | avg. block | median block | % missing | avg. block | median block |
| PEMS-BAY | 0.02 | 12.0 | 12.0 | 25.00 | 3.33 | 3.0 |
| METR-LA | 8.10 | 12.44 | 9.0 | 23.00 | 2.22 | 3.0 |

Additionally, it's worth noting that PEMS-BAY contains data spanning 6 months, while METR-LA covers a period of 4 months, with both datasets having a 5-minute sampling rate. In terms of implementation, we employ a horizon window approach with a horizon of 24 steps, equivalent to 2 hours of data. We utilize an adjacency matrix generated from a thresholded Gaussian kernel applied to geographical distances. The data is divided into train, validation, and test sets, with proportions of 70%, 10%, and 20%, respectively. Everything described in this paragraph is consistent with Cini et al. (2022),

## A.6 SENSITIVITY ANALYSIS

In this section, we present a sensitivity analysis to examine how varying amounts of missing values affect the performance of the proposed model. It is important to note that, to ensure comparability with the experimentation conducted by Cini et al. (2022), the model was initially trained on a dataset where 60% of the samples in each batch were randomly removed. Subsequently, the model was tested by running it five times on datasets with varying amounts of missing values.

Furthermore, it is worth mentioning that the model was trained using the hyperparameters established in Appendix A.2, which were optimized for a 23% missing value scenario. This deviation from the original training setup may impact the results that can be seen in Section 5.2 and potentially lead to decreased performance.

Lastly, training a model to impute data with a certain amount of missing values and evaluating it under different conditions introduces a distribution shift, which can be detrimental to our proposed model as it violates the i.i.d. assumption established in Section 3.1.

Table 7: Results of the sensitivity analysis, measured in terms of MAE averaged over 5 runs

| METR-LA | | | | | | | | | |
|---|---|---|---|---|---|---|---|---|---|
| % Missing | 10 | 20 | 30 | 40 | 50 | 60 | 70 | 80 | 90 |
| GRIN | $1.87_{\pm 0.01}$ | $1.90_{\pm 0.00}$ | $1.94_{\pm 0.00}$ | $1.98_{\pm 0.00}$ | $2.04_{\pm 0.00}$ | $2.11_{\pm 0.00}$ | $2.22_{\pm 0.00}$ | $2.40_{\pm 0.00}$ | $2.84_{\pm 0.00}$ |
| BRITS | $2.32_{\pm 0.01}$ | $2.34_{\pm 0.00}$ | $2.36_{\pm 0.00}$ | $2.40_{\pm 0.00}$ | $2.47_{\pm 0.00}$ | $2.57_{\pm 0.01}$ | $2.76_{\pm 0.00}$ | $3.08_{\pm 0.00}$ | $4.02_{\pm 0.01}$ |
| G-TIGRE | $2.28_{\pm 0.03}$ | $2.30_{\pm 0.02}$ | $2.35_{\pm 0.04}$ | $2.39_{\pm 0.02}$ | $2.47_{\pm 0.01}$ | $2.62_{\pm 0.03}$ | $2.86_{\pm 0.04}$ | $3.64_{\pm 0.04}$ | $5.42_{\pm 0.06}$ |

The results are presented in both Table 7 and Figure 3. It is evident that G-TIGRE maintains a competitive position, closely trailing BRITS, until reaching a 50% missing value rate, where both models perform similarly. However, G-TIGRE starts to degrade more rapidly beyond this point, with significantly deteriorating results when faced with 80% missing values.

These results provide further insights into the behaviour and limitations of our proposed model, highlighting that G-TIGRE is particularly susceptible to distribution drift. Also, it suggests that G-TIGRE may not be the most suitable choice for extreme scenarios with more than 80% missing values.

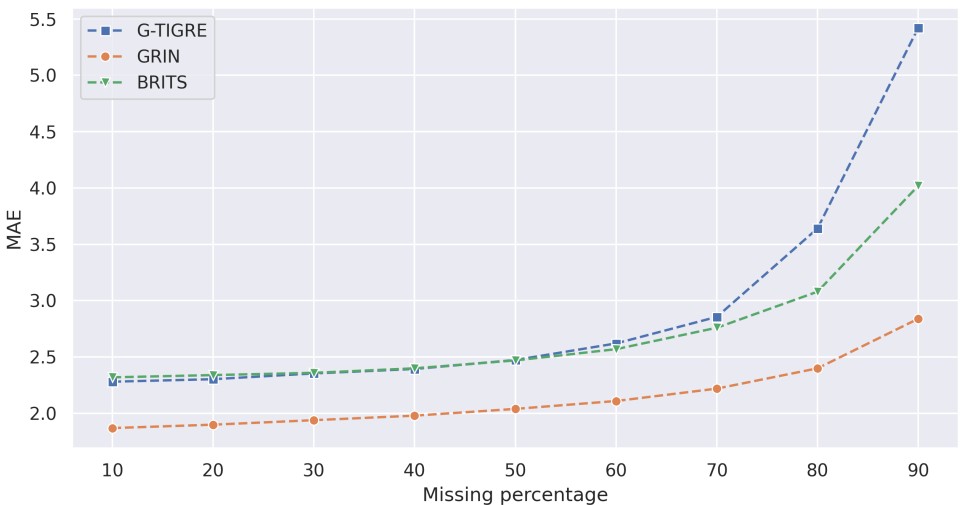

Figure 3: A graphical representation of the sensitivity analysis results presented in Table 7.

