# OpenReview forum: "G-TIGRE: A new generative framework for Multivariate Time Series Imputation By Graph Neural Networks"
_ICLR.cc/2024/Conference — ICLR 2024 Conference Withdrawn Submission_

### Official Review · Reviewer_FuJh · 2023-10-26

**Soundness:** 2 fair
**Presentation:** 3 good
**Contribution:** 1 poor
**Rating:** 3
**Confidence:** 4

**Summary:**

The paper introduces a methodology for training a standard (spatio-temporal) GNN architecture to perform time series imputation, exploiting adversarial training. Although the paper is well-written and the method sound, some claims are questionable, empirical performance is disappointing (up to 50% worse w.r.t. the state of the art) and the novelty is quite limited.

**Strengths:**

* The introduced approach is sound.
* GNN-based probabilistic imputation is underexplored.
* Good presentation.

**Weaknesses:**

The major weaknesses of the paper are the limited novelty and the poor empirical results.

* The proposal is that of training a simple STGNN (adapted from a standard baseline) with the GAN framework to perform imputation, following [1]. Adding an adversarial loss to an existing method might not be novel enough for acceptance at ICLR, especially if the empirical results are weak.
* There are several methods that exploit modern generative models for time series imputation, e.g., diffusion models [2], making novelty even more limited.
* There are a few unjustified claims. For example, G-TIGRE is trained to match the data distribution w.r.t. the training data, so it cannot handle non-stationarity (i.e., distribution shifts at inference time) as claimed in the paper.
* The empirical analysis is quite limited as it consists of just 2 benchmarks and, e.g., does not assess the calibration of the probabilistic imputation model. Furthermore, the reconstruction accuracy is quite poor (up to 50% worse than a popular baseline of 2 years ago). More up-to-date baselines have not been considered, e.g., [3].
* Finally, the improvement w.r.t. the non-generative baseline in the ablation study seems not statistically significant by looking at the standard deviations. This makes me question the effectiveness of the method.

Given the above, I cannot recommend acceptance.

[1] Yoon et al. "GAIN: Missing data imputation using generative adversarial nets." ICML 2018

[2] Tashiro et al. "CSDI: Conditional score-based diffusion models for probabilistic time series imputation." NeurIPS 2021

[3] Marisca et al. "Learning to Reconstruct Missing Data from Spatiotemporal Graphs with Sparse Observations" NeurIPS 2022

**Questions:**

- I don't understand how G-TIGRE could handle non-stationary data.

**Details Of Ethics Concerns:**

--

---

### Official Review · Reviewer_vBVz · 2023-10-26

**Soundness:** 2 fair
**Presentation:** 2 fair
**Contribution:** 2 fair
**Rating:** 3
**Confidence:** 3

**Summary:**

The authors study the taks of missing value imputation in multivariate time series in the case where graphs are available. The approach proposed relies on two ingredients: Graph Convolutional Networks (GCNs) and Generative Adversarial Networks (GANs). By considering these two methods and wrapping in the in a single approach the authors show that this approach provides competitive performance to the state of the art.

**Strengths:**

The authors consider the task of missing value imputation in multivariate time series, by making the most of information available through graphs, and considering as well lateral tooling like GANs. The idea of merging this approach is interesting on its own and essentially opens up a new path of potential approaches. In this sense, the main strength of the paper is the proposition of joining these two paradigms into a single model to provide a new way to solve the problem at hand.

**Weaknesses:**

The main weakness of the paper in the current state is that it seems that the authors are taking existing elements in the literature and putting them together, leaving novelty to be rather marginal. Further, the performance shown by the proposed method seems to be competitive to the state of the art, and it is unclear what could be necessary to provide the performance improvement required to perform best.

Further, there is no analysis showing what the time execution comparison to the state of the art, together with the memory/computational requirements for this approaches. Is the proposed approach faster? Is it more scalable? Is there any particular regime where this approach performs better? Is there any cross effect of the sparsity of the graphs and the amount of missing values in the time series?

A point to improve in the paper is writing. Whereas the authors have a clearly structured and clear writing style, the main paper is not self-contained. Several variables in the main paper are defined in the supp. material and the authors even refer equations parameters in the main paper to the supp. material. In this way one can not fully grasp the mathematical essence of the paper without the supp. material. I would suggest the authors to fix this.

**Questions:**

- I guess the binary complement of $M_t$ is $1-M_t$. Is this correct?

- Typo on top of page 4: citing: "the objective is from $\widehat{X_t}$ build $\widehat{X_t}$".
After eq.1 there is a hint of what the authors meant: "$\hat{x}_h^i$ is reconstructed version of $\tilde{x}_h^i$". Is this correct?

- Why did the authors use the symbol $\odot$ for the dot product? Maybe I am wrong, but I believe such a symbol is standard of other products, like the Hadamard product: https://en.wikipedia.org/wiki/Hadamard_product_(matrices).
This notation choice is highly confusing. I would suggest the authors to find another notation for this. What about the tranditional symbol $< , > $ ?

- I would still follow the above suggestion to the definition of graphs that the authors have. In general, a graph is defined as a pair of a set of Vertices and Edges, for which traditionally one uses $G = (V, E)$. I guess a similar notation can be used, instead of $<,>$. Unless the authors consider that a graph as in this context is some sort of a inner product.

See notation for inner product (https://en.wikipedia.org/wiki/Dot_product), dot product (https://en.wikipedia.org/wiki/Dot_product), and graphs (https://en.wikipedia.org/wiki/Graph_theory)

- One of the key elements in the paper is the description of losses. In the case of eq. 2, where a particular loss is introduced, it is completely relying to notation introduced in the appendix. This makes the main paper not self contained even at the level of notation. I would suggest the authors to address modifications so that the main paper is self-contained. Of course the supplementary material can be used for clarifications, but the main paper should be self contained at least at the level of notation.

- Further, the authors are making latex references to equations in the supplementary material, making the main paper not self-contained.

In the last paragraph of p.6 one needs to go to the supp. material to understand the definition of variables in equations. Further, the authors again introduce the standard dot product, which was done already at the beginning of the paper.

- Notation in appendix: initially in the main paper it was stated that $x_t^i$ was a vector, but in equation 8 it seems to be a scalar.

---

### Official Review · Reviewer_mn2X · 2023-11-01

**Soundness:** 2 fair
**Presentation:** 3 good
**Contribution:** 2 fair
**Rating:** 3
**Confidence:** 3

**Summary:**

This paper proposes a new method imputation in multivariate time-series, based on GNNs and GANs.

The proposed model, G-TIGRE, is a form of GAN, with the generator learning an imputed version of the input matrix and the discriminator learning a mask distinguishing real from imputed data. The generative architecture relies on an RNN to extract temporal representations, then GCN layers to extract a latent graph (capturing the ‘spatial’ relationship between temporal representations, assumed fixed over time). Both the generative and the discriminative models are bidirectional, consisting of two models for forward/backwards time direction which are then merged into an MLP.

Experimental results include two traffic datasets.

**Strengths:**

- Imputation in time-series is an important and challenging problem. Addressing it with generative methods is sensible, and GNNs have shown promise in time-series modelling -- so the idea of combining them is well-motivated.

- The paper is clear and quite easy to follow.

- Initial experimental results are promising and useful ablations are provided.

**Weaknesses:**

- The experimental investigation is too limited to validate the usefulness of G-TIGRE. We don't necessarily need to see top-of-the board performance but the method is only studied on two similar datasets, when many other imputation benchmarks exist (e.g.  PhysioNet 2012, Beijing Air-Quality, UCI Electricity datasets, etc.). As a result, experimental results and ablations remain unconvincing that the proposed method has real added value. Example:
  - Results with/without adversarial loss overlap. So the adversarial framework doesn't help that much... What insights do authors propose on this/to justify its value? perhaps the more stable autoencoder structure is equally promising here.
  - Same comment/question about results with the fully connected graph.

- Some additional references:

Yonghong Luo, Xiangrui Cai, Ying ZHANG, Jun Xu, and Yuan xiaojie. Multivariate time series imputation with generative adversarial networks. Advances in Neural Information Processing Systems, 2018.

Yonghong Luo, Ying Zhang, Xiangrui Cai, and Xiaojie Yuan. E2GAN: End-to-end generative adversarial network for multivariate time series imputation. Proceedings of the Twenty-Eighth International Joint Conference on Artificial Intelligence, 2019.

- Presentation in Sec 4.1. needs improving: authors should not expect the reader to have read the appendix to understand the method being proposed. If these elements are necessary, include them in the main text. In particular:
  - please define $m,  \hat{m}, b$ in the main text
  - "However, as in the last citation, the objective function to be minimized by G differs slightly from Equation 11." see comment about the reader not having to read the appendix to understand the method.
  - Discriminator loss not given in the main text. Is it Equation (2)? It is not referred to in the text.


Minor:
- "we observe a graph $G_t$ with $N_t$ at each time step": missing word?
- this should be rephrased: "our approach imputes missing values in the context of i.i.d."

**Questions:**

- Why not use a transformer architecture instead of RNNs to encode temporal information?
- Does this method also help when the time-series is not necessarily generated from a graph model? More diversity in the tasks being studied would help the paper (see comment above).
- If GRIN has a stationarity constraint that G-TIGRE relaxes, are the dataset studied stationary? Would a non-stationary dataset allow to better showcase the added value of G-TIGRE?
- Authors claim G-TIGRE "demonstrates enhanced flexibility". what does flexibility mean? where is this demonstrated?

---

### Official Review · Reviewer_XP5p · 2023-11-04

**Soundness:** 2 fair
**Presentation:** 2 fair
**Contribution:** 2 fair
**Rating:** 3
**Confidence:** 4

**Summary:**

The paper proposes to combine graph neural networks with generative adversarial networks (GANs) to develop a multivariate time-series imputation method.

While paper considers an important task, the literature review seems to fall short, and does not consider recent developments in the area. In terms of writing, the reliance on Cini et al. 2022 makes the paper not self-contained. As a result, it is difficult to evaluate the merits in light of contemporary methods.

**Strengths:**

1. The paper aims to solve an important task.
2. The paper is well motivated.

**Weaknesses:**

1. The paper is not self-contained. To analyze the results in Table 1 the paper points the readers to Cini et al. 2022. While it is okay to follow a paper's evaluation methodology, it is unclear which works these baselines come from. Moreover, none of these baselines are described or put into the context. This makes it difficult to see to why other methods did not perform well, and how the proposed method overcomes these pitfalls.

2. The paper does not consider recent relevant such as those which leverage transformers for missing value imputation [1], generative modelling [2], and diffusion models [3]. The literature review also seems completely based on Cini et al. 2022, while there have been significant efforts which seem to be excluded from the discussion. I recommend a thorough literature review of spatio-temporal imputation and a re-write of the related works and adding more baselines based on the search.

[1] Bansal et al.,  "Missing Value Imputation on Multidimensional Time Series." In VLDB. 2021.
[2] Kim et al., Probabilistic Imputation for Time-series Classification with Missing Data, ICML 2023.
[3] Yun et al., "Imputation as Inpainting: Diffusion models for SpatioTemporal Data Imputation." ICLR 2023.

**Questions:**

See Weaknesses.